# Regular Periodic Surface Structures on Indium Tin Oxide Film Efficiently Fabricated by Femtosecond Laser Direct Writing with a Cylindrical Lens

**DOI:** 10.3390/ma15155092

**Published:** 2022-07-22

**Authors:** Long Chen, Jian Yang, Qilin Jiang, Kaiqiang Cao, Jukun Liu, Tianqing Jia, Zhenrong Sun, Hongxing Xu

**Affiliations:** 1State Key Laboratory of Precision Spectroscopy, School of Physics and Electronic Science, East China Normal University, Shanghai 200062, China; lchen@phy.ecnu.edu.cn (L.C.); jyang@lps.ecnu.edu.cn (J.Y.); jiangqilin@tju.edu.cn (Q.J.); caokq1010@163.com (K.C.); zrsun@phy.ecnu.edu.cn (Z.S.); hxxu@whu.edu.cn (H.X.); 2College of Science, Shanghai Institute of Technology, Shanghai 201418, China

**Keywords:** laser-induced periodic surface structures, femtosecond laser direct writing, ITO film, structural coloring, cylindrical lens

## Abstract

Regular laser-induced periodic surface structures (LIPSS) were efficiently fabricated on indium tin oxide (ITO) films by femtosecond laser direct writing with a cylindrical lens. It was found that randomly distributed nanoparticles and high spatial frequency LIPSSs (HSFL) formed on the surface after a small number of cumulative incident laser pulses per spot, and regular low spatial frequency LIPSSs (LSFL) appeared when more laser pulses accumulated. The mechanism of the transition was studied by real-time absorptance measurement and theoretical simulation. Results show that the interference between incident laser and surface plasmon polaritons (SPPs) excited by random surface scatterers facilitates the formation of prototype LSFLs, which in turn enhances light absorption and SPP excitation following laser pulses. The effects of scanning velocity and laser fluence on LSFL quality were discussed in detail. Moreover, large-area extremely regular LSFL with a diameter of 30 mm were efficiently fabricated on an ITO film by femtosecond laser direct writing with the cylindrical lens. The fabricated LSFLs on the ITO film demonstrate vivid structural color. During LSFL processing, the decrease of ITO film thickness leads to the increase of near-infrared optical transmittance.

## 1. Introduction

Ultrafast laser has become an effective tool for micro/nano-processing [1,2,3,4,5]. Versatile methods are offered for cutting, drilling and modification of different kinds of materials [6,7,8]. In the large variety of subfields related to ultrafast laser micro/nano-processing, laser-induced periodic surface structures (LIPSS) have received considerable attention over the past decades [9,10]. The formation of LIPSSs is a universal phenomenon observed on semiconductors, dielectrics, metals and thin films [2,5,11,12,13,14,15,16,17,18]. According to the relationship between incident laser wavelength *λ* and the period of induced surface structures *Λ*, LIPSS can be classified as high spatial frequency LIPSS (HSFL, *Λ* < 0.5*λ*) and low spatial frequency LIPSS (LSFL, 0.5*λ* < *Λ* < *λ*) [5,18,19,20,21,22,23,24,25,26]. LIPSSs exhibit great potential for applications in many fields, such as structural colors [18,27,28,29], color-based anti-counterfeiting [30], superhydrophobicity [31,32] and birefringence [33,34,35,36].

Indium-tin oxide (ITO) films have a wide bandgap of 3.5–4.3 eV. As a semiconductor, ITO has high carrier concentration, low electrical resistivity and high optical transparency at visible wavelengths [37,38], and is widely used as a transparent electrode in organic light emitting devices (OLED), liquid crystal displays, thin film solar cells, etc. [39,40,41,42,43,44]. However, the transmittance of untreated ITO film in the infrared band is low, which will significantly reduce the light absorption efficiency of thin-film solar cells [45,46] and limit the application of ITO film as a transparent electrode in the infrared band.

Nanostructures fabricated on the ITO film can effectively modulate its electric and optical properties, especially surface resistance and optical transmittance [10,16,41,47,48,49,50,51], and can be used for application in film solar cells and OLED devices. Reinhardt et al. enhanced the robustness of an ITO film in a severe environment (i.e., strong acid) by fabricating LIPSS with a nanosecond laser, and the incorporation of silicon into ITO is considered to be the reason for the robustness of this sub-pattern against acidic environments [52]. Liu et al. reported that LIPSSs were processed on ITO film by picosecond laser, which ensured low resistance and improved IR transmittance [53]. At present, the efficient fabrication of large-area regular nanostructures on ITO films has attracted great attention.

In this paper, regular LSFLs on ITO thin films were efficiently fabricated by femtosecond laser direct writing with a cylindrical lens. It was found that the number of cumulative laser pulses per spot played a crucial role in LSFL formation. Unlike semiconductor and metal surfaces, such as silicon and copper, randomly distributed nanoparticles and HSFL parallel to the incident laser polarization appeared on the surface with only a few incident laser pulses. With more incident laser pulses accumulated, surface geometry turned into regular LSFLs. Through real-time absorptance measurement and theoretical simulation, it was found that the formation of regular LSFLs was caused by the interference between incident light and surface plasmon polaritons (SPPs) excited by random surface scatterers. Large-area extremely regular LSFLs were efficiently fabricated on an ITO film with a diameter of 30 mm by a femtosecond laser focused with a cylindrical lens. The linear focusing of the cylindrical lens greatly improves the processing efficiency, and the LIPSSs fabricated are extremely regular. The fabricated LSFLs on the ITO film demonstrate great potential in structural color. During LSFL processing, the decrease in ITO film thickness leads to an increase in near-infrared optical transmittance.

## 2. Materials and Methods

### 2.1. Laser Direct Writing Setup and Sample Characterization

Figure 1a shows the experimental setup for laser direct writing. A commercial femtosecond laser (Light Conversion Ltd., Vilnius, Lithuania) was used in the experiment, which produced laser pulses of wavelength 1030 nm, pulse width 250 fs and pulse energy 1 mJ with a repetition rate of 1 kHz. The laser power and polarization were adjusted through a combination of a half-wave plate and a Glan prism. A mechanical shutter was used to control the laser irradiation time. The femtosecond laser was incident vertically on the sample through a cylindrical lens with a focal length of 50 mm. The focal spot was 15 µm wide along the minor axis (1/*e*^2^ intensity) and 4.0 mm long along the major axis. Compared with a common circular lens with the same focal length (focal spot diameter of 15 μm), the fabrication efficiency is 266 times higher. Moreover, the LIPSSs produced with a cylindrical lens are also more regular [29,36]. An electronically controlled half-wave plate was used to continuously rotate the direction of laser polarization. A colinear monitoring system consisting of a white light source, a dichroic mirror and a charge-coupled device (CCD) were built to monitor the femtosecond laser direct writing in real-time. Figure 1b shows the schematic of the laser polarization, focal spot and scanning direction.

A glass substrate with a diameter of 30 mm coated with ITO film of thickness 175 ± 10 nm (MTI-group, Jiangsu, China) was used in the experiments. The sample was mounted on an x/y/z/θ translation stage. After femtosecond laser direct writing, the samples were cleaned in an ultrasonic bath with deionized water, isopropanol and, finally, acetone [16]. The surface morphology was examined using a scanning electron microscope (SEM, Sigma 300, ZEISS, Oberkochen, Germany) and an optical confocal microscope (Smartproof 5, ZEISS, Oberkochen, Germany). The resolution of the optical confocal microscope was 5 nm on the Z axis and 100 nm on the X/Y axis. Semi-quantitative measurement of the chemical composition was performed using energy-dispersive X-ray spectroscopy (EDS) (SEM, Sigma 300, ZEISS, Oberkochen, Germany). The EDS instrument was an accessory of the SEM and had an information depth of 600 nm. The transmittance spectra of the ITO film were measured using an ultravioletivisible-NIR spectrophotometer (LAMBDA 950, PerkinElmer, Waltham, MA, USA).

### 2.2. Real-Time Absorptance Measurement Setup

The laser was focused onto the surface of the sample at an inclination angle of 70° (the angle between the incident laser and the sample surface). The laser fluence and polarization were adjusted through a combination of a half-wave plate and a Glan prism. Two energy meters were placed on the reflected and transmitted light path to measure the reflected and transmitted energy of each laser pulse in real-time. The absorptance *A* could be calculated as
(1)A=1−R−T,
where *R* is reflectance and *T* is transmittance. The reflectance and transmittance could be calculated by taking the ratio between the measured and the incident pulse energies.

The transmitted and reflected lasers were focused on the energy meter through a quartz lens with a focal length of 50 mm and a diameter of 75 mm in order to collect all the laser energy scattered and diffracted by the sample structure. The lens had a collection angle of 70° (the corresponding collection angle of the reflected light was in the range of 35−105°). The scattered light beyond the collection angle was also measured using an energy meter, but its energy was too small and below the energy meter measurement range. Therefore, most of the scattered light has been included in the measurement of absorption.

The real-time reflectance measurement should be carried out at an appropriate angle. For a normal incident laser, the detector cannot directly receive the reflected light in real-time. Therefore, an incident angle of 70° was set for the absorption measurement, and the detector could receive the reflected light well.

It is known that the LIPSS periods vary with the incident angle [54,55,56]. However, the incident angle of 70° was very close to 90°, and the periods of the LIPSSs were only 50 nm different from those when using a normal incident laser [56]. The periods of the two cases were very close, and the LSFL formation processes were basically the same. Therefore, it is feasible to use an incident angle of 70° to replace normal incidence in the absorption measurement during LIPSS formation.

### 2.3. Structural Color Measurement Setup

Figure 2 shows a schematic diagram of the measurement setup to characterize the structural color of the samples from different viewing angles [18,29]. A wide spectral light source (a tungsten halogen lamp with a spectral range of 400–2200 nm) illuminated vertically on the sample surface, and structural colors were measured at different viewing angles using a CCD moving in the Y-Z plane (perpendicular to the LIPSS direction). The angle between the CCD and the Z axis was defined as the observation angle *α*. The diffraction equation reveals the relationship between the diffraction wavelength *λ* and the observation angle *α*:(2)nλ=dsinα,
where the integer *n* is the diffraction order and *d* is the period of LIPSS.

## 3. Results and Discussion

### 3.1. Formation of LSFL on ITO Film

The formation process of regular LSFLs on the ITO film is shown in Figure 3, in which the laser scanning velocity decreased from 12 to 3 mm/s with the pulse fluence fixed at 0.66 J/cm^2^. Large differences can be observed across different scanning velocities. When the scanning velocity was 12 mm/s, the average number of cumulative pulses per spot was only 1.2. As shown in Figure 3a, slight ablation appeared on the ITO film with some randomly distributed nanoparticles. Due to the high scanning velocity, loosely spaced burnt regions were formed. When the scanning velocity decreased to 6 mm/s, the number of cumulative pulses increased to 2.4 on average. The original random nanoparticles evolved into many short and irregular HSFLs parallel to the laser polarization (Figure 3b). Figure 3c shows the 2D Fourier transform (FT) images of the SEM image of Figure 3b. Figure 3g,h are the corresponding FT spectrums for k_x_ = 0 μm^−1^ and k_y_ = 0 μm^−1^ in Figure 3c, respectively. As shown in Figure 3g, the peak of the FT spectrum was 3.0 ± 1 μm^−1^, which indicated that the HSFL period was 330 ± 100 nm. Meanwhile, some fuzzy prototype LSFLs perpendicular to the laser polarization were formed. As shown in Figure 3h, the peak of the FT spectrum was 1.5 ± 0.4 μm^−1^, which indicated that the LSFL period was 660 ± 150 nm. The LSFL produced was very irregular and had a wide range of periodic fluctuations.

It was reported that HSFLs parallel to the laser polarization were formed on the LSFL ridge of the metal surface, which was attributed to the cavitation instability in the molten surface layer [57,58,59].

When the scanning velocity further decreased to 4 mm/s, the LSFL perpendicular to the laser polarization direction became clearer and more regular (Figure 3d). Only short HSFLs appeared on the stripes of the LSFL. However, since the number of accumulated pulses per spot was still less than 4, the LSFLs were rather curved and irregular. Finally, when the scanning velocity decreased to 3 mm/s, very regular LSFLs formed on the surface as shown in Figure 3e. Figure 3f depicts the 2D Fourier transform (FT) image of the LSFLs and Figure 3i is the corresponding FT spectrum for k_y_ = 0 μm^−1^. As shown in Figure 3f, the peak of the FT spectrum was 1.075 ± 0.008 μm^−1^, which indicated that the LSFL period was 930 ± 5 nm.

Since the scanning velocity directly determines the number of cumulative laser pulses per spot, it is helpful to examine the optical response of the ITO film for each consecutive laser pulse. Figure 4 shows the measured reflectance, transmittance and absorptance of the ITO film irradiated by 20 consecutive laser pulses with the pulse fluence fixed at 0.66 J/cm^2^. For the first laser pulse radiation, the reflectance and transmittance of the ITO film were 20% and 62% respectively, corresponding to a very low absorptance value of only 18%. The absorptance was enhanced rapidly to 32% of the radiation by the 5th laser pulse. However, the absorptance began to decrease with further laser pulse radiation. It was only 15% of the radiation by the 20th laser pulse, which indicated that most of the ITO film had been ablated off.

Under femtosecond laser irradiation, ITO was rapidly excited to form a surface plasma layer, and the properties of the material changed from dielectric to metallic. After the initial laser pulse irradiation, periodic ablation stripes covering random defects were formed on the ITO film, as shown in Figure 3a [18]. These ablation stripes and random defects would lead to the uneven absorption of subsequent laser pulses on ITO film, resulting in an irregular distribution of nanoparticles and HSFLs. These surface patterns served as scattering sites for the following laser pulses, and surface plasmon polarizations (SPPs) could be excited on the surface [5,18,19,25,60], which is studied in detail in Section 3.2 and the corresponding descriptions. The interference between the incident laser and SPPs led to the periodic distribution of energy and electronic temperature. The electrons then transferred the energy to the crystal lattice through electro-phonon coupling, which caused the ablation of the material and the formation of prototype LSFLs [5,11,25,60]. The appearance of LSFLs due to SPPs further enhanced light absorption and SPP excitation for subsequent laser pulse irradiation, causing the enhanced absorption for following pulses (Figure 4b). This led to an absolute advantage of periodic light absorption compared with the original non-uniform absorption [19], forming a self-enhancing process. Therefore, for a lower scanning velocity, more pulses accumulated on a single spot and the LSFL became straighter and more regular.

### 3.2. Theoretical Simulation of Light Field Distribution

To further validate the LSFL formation mechanism we proposed, COMSOL Multiphysics software was used to calculate the electric field distribution by solving Maxwell’s equations. Figure 5 presents the schematic of the simulation setup [61,62]. The simulation domain was 15 × 10 × 0.875 μm^3^ in volume, where four layers of air (0.5 μm), the ITO film in the excited states (0.075 μm), ground states (0.1 μm) and the glass layer (0.2 μm) were enclosed. The dielectric permittivity of ITO in the ground states is 3.6159 + 0.0085i, which is taken from reference [63]. The dielectric permittivity of ITO in the excited states is calculated to be −4.45 + 0.35i according to the Drude model, in which the carrier density is calculated from the Boltzmann’s transport equation [25,64]. The details are shown in the Appendix A.

Figure 3a,b show that random defects and irregular HSFLs formed on the ITO film with high scanning velocity. The sizes of the HSFLs are 500 × 100 nm^2^, with orientation parallel to the laser polarization. The depths of the HSFLs were measured to be in the range of several tens to hundreds of nanometers [65,66,67]. In order to simplify the simulation, the ablated regions were represented by rectangular grooves with sizes of 500 × 100 × 75 nm^3^. The selection of other sizes such as 600 × 120 × 60 nm^3^ will not have a great influence on the simulation results. The positions and numbers of these rectangular grooves are randomly distributed according to the experimental results. However, the orientation changes after being irradiated by a series of laser pulses.

Plane waves irradiated the sample in the normal direction. The boundary conditions in the X directions were periodic boundary (PBC) conditions, while in other directions scattering boundary conditions (SBC) were implemented [61,62].

We first simulated the light field distribution on the upper surface irradiated by the initial laser pulse, in which HSFsL parallel to the laser polarization were formed. As shown in Figure 6a, the magnified image of the black square region shows nanogrooves with orientations parallel to the laser polarization. Some periodic distribution of the light field occurred on the surface as a result of the interference between the incident light and SPPs excited by the HSFL nanostructures. The 2D FT image in Figure 6d shows that periodicity parallel to the laser polarization was stronger than that in the vertical direction. These light intensity distributions would lead to prototype LSFLs perpendicular to the laser polarization. In order to further analyze the evolution of light intensity distribution caused by the formation of the prototype LSFLs, the grooves were placed separately at different angles from the polarization direction, as shown in Figure 6b. The 2DFT image in Figure 6e shows that periodicity parallel to the laser polarization was obviously reduced and became lower than that in the vertical direction. The FT spectrum in k_x_ direction was 1.43 ± 0.35 μm^−1^, corresponding to a period of 700 ± 160 nm. The ripple-distributed light intensity was significantly enhanced, in agreement with the discussion in the previous section that the formation of LSFL will further enhance SPP excitation. With the groove directions completely perpendicular to the polarization direction as shown in Figure 6c, the light intensity distribution was very regular with a period of 930 ± 15 nm (Figure 6f), and the periodicity parallel to the laser polarization almost completely disappeared.

### 3.3. Effects of Scanning Velocity and Laser Fluence on LSFL Quality

The scanning velocity and laser fluence have a great influence on the LSFL quality, so a detailed study of the effects of these two parameters is necessary for the fabrication of desirable LSFLs. Figure 7 and Figure 8 show the SEM images and ripple depth of LSFLs fabricated at different scanning velocities from 4 to 1 mm/s. As shown in Figure 7a, when the laser scanning velocity was 4 mm/s, the number of cumulative laser pulses per spot was only 3.5 on average. The LSFLs formed on the surface show fuzzy and irregular shapes with an average depth of 12 nm. When the scanning velocity was reduced to 3 mm/s, very regular LSFLs appeared as shown in Figure 7b with the FT spectrum peak at 1.075 ± 0.008 μm^−1^ corresponding to a LIPSS period of 930 ± 5 nm. Figure 7e shows the depth profile along the red line in Figure 7b. The depth of each ripple fluctuated in a very small range, which shows that the LSFLs are very regular. The average depth reached a maximum value of 48 nm. Further decrease in the scanning velocity results in the LSFLs being ablated off (Figure 7c,d). When the scanning velocity was reduced to 1 mm/s, the LSFLs almost disappeared. Very thin and shallow ripples (10 nm) appeared on the surface of the glass substrate.

The main components of the ITO film are indium, tin and oxygen. Therefore, the weight percentage of indium ions can be used to characterize the removal of the ITO film. Figure 9a–h shows the SEM images of the geometry and the distribution of the indium ion content for the LSFL fabricated under scanning velocities from 4 to 1 mm/s with the pulse fluence fixed at 0.66 J/cm^2^. When the scanning velocity was 4 mm/s, as shown in Figure 9e, the indium ion content was relatively high, reaching 39.3%, indicating that the main component of the surface was ITO. When the scanning velocity was reduced to 3 mm/s, very regular LSFLs formed and the indium ions were evenly distributed in the vertical direction (Figure 9b,f). Figure 9k shows the distribution of indium ions along the red line in (b) and the indium ion content distribution was very regular. The whole ITO film was gradually ablated off with further decreases in the scanning velocity. With a scanning velocity of 1 mm/s, the indium ion content on the surface was reduced to merely 2.42% as shown in Figure 9h,j. 

Figure 10 and Figure 11 show the SEM images and ripple depth of the LSFLs fabricated on the ITO film for different pulse fluences at a fixed scanning velocity of 3 mm/s. As shown in Figure 10a, when the laser fluence was 0.55 J/cm^2^, sprototype LSFLs formed on the ITO film surface. Due to the low laser fluence, the ITO film in the gap regions was not completely removed and the formed LSFLs were very irregular. There were some HSFL patterns parallel to the laser polarization direction. The average depth measured by a confocal microscope was 24 nm, as shown in Figure 11. When the laser fluence increased to 0.66 J/cm^2^, very regular LSFLs with a period of 930 ± 5 nm were formed. Figure 11 shows that the depth reached a maximum value of 48 nm at this pulse fluence. With further increasea in the pulse fluence, LSFLa began to be ablated off. When the pulse fluence increased to 0.89 J/cm^2^, LSFL almost disappeared, and the depth of the ripples was only 9 nm.

In order to better study the formation process of LSFLa, we studied different types of micro/nanostructures depending on the laser fluence and scanning velocity, as shown in Figure 12. When the scanning velocity was in the range of 3–4 mm/s and the laser fluence was in the range of 0.64–0.79 J/cm^2^, regular LSFLs covered the entire ablation area. When the laser fluence exceeded 1.3 J/cm^2^, LSFLs were ablated off. When the laser fluence was lower than 0.3 J/cm^2^, loosely spaced burnt regions were formed on the ITO film.

The formation of LSFLs on ITO films with different thicknesses (such as 50 nm, 175 nm and 600 nm) was also studied. LSFLs can be prepared on all these ITO films with different thicknesses, but the quality varies greatly. For ITO films with a thickness of 50 nm, they were easily completely ablated before regular LSFLs formed. While for ITO films with a thickness of 600 nm, the laser fluence required for processing LSFLs was higher than 1.0 J/cm^2^ at a scanning velocity of 3 mm/s. The heat accumulation during laser processing was very serious, which resulted in irregular LSFLs [5,52,53,68]. ITO film with a thickness of 175 nm was suitable for processing LSFLs with extreme regularity.

## 4. Large-Area LSFL Fabrication and Applications

### 4.1. Fabrication of Large-Area LSFL

Based on the study of the LSFL formation mechanism and effects of the fabrication parameters, large-area extremely regular LSFLs with a diameter of 30 mm were prepared efficiently on the ITO thin film by femtosecond laser direct writing with a cylindrical lens [29]. The laser fluence was 0.66 J/cm^2^ and scanning velocity was 3.0 mm/s, indicating that the interval between adjacent scanning lines was 1.5 mm and the number of cumulative pulses per spot was about five. As shown in Figure 13a, the LSFLs were extremely straight and regular with a period of 930 nm. Figure 13b shows the optical image of the LSFLs prepared. Different colors due to diffraction can be clearly observed. Note that large-area LSFLs on the ITO film with a diameter of 30 mm can be fabricated very efficiently within 4 min using cylindrical lens focusing.

### 4.2. Structural Color Patterning

Various patterns of LSFLs were fabricated by placing masks of different shapes on the ITO sample and the measurement setup shown in Figure 2 was used to characterize the structural color from different viewing angles [29]. Due to the diffraction of the uniform LSFLs, different structural colors could be displayed on the CCD by observing from different angles, as shown in Figure 14a–i. Figure 14j shows the diffraction spectra of the pattern composed of extremely regular LSFLs with peaks at 470, 540 and 660 nm, where the full widths at half-maximum (FWHM) are 32.7, 30.6 and 35.2 nm, respectively. Each diffraction spectrum has only one peak, and the FWHM values are all less than 36 nm. Therefore, all the structural colors were very bright and pure because the fabricated LSFLs were extremely straight and regular.

### 4.3. Increasing Near-Infrared Optical Transmittance

Figure 15 shows light transmittance spectra of ITO films with LSFLs in the wavelength range of 300–2000 nm. The transmittance only changed slightly with laser fluences in the wavelength range of 300–1000 nm. In the near-infrared region of 1200–1900 nm, the average transmittance of the original ITO film was 37.96% in Table 1. With femtosecond laser direct writing of LSFLs, the transmittance in the near-infrared region increased. When the pulse fluence was 0.66 J/cm^2^, very regular LSFLs formed, and the average transmittance increased to 71.3%. When the fluence further increased to 0.89 J/cm^2^, the ITO film was almost completely removed, and the transmittance increased to 91.0%. The near-infrared optical properties of the ITO film can be tuned by femtosecond laser writing of LSFLs at different fabrication parameters. The increase in transmittance was mainly caused by the decrease in ITO film thickness. LSFL processing is accompanied by material removal and a corresponding increase in transmittance.

## 5. Conclusions

In this paper, regular LSFLs were efficiently prepared on ITO films by femtosecond laser direct writing with a cylindrical lens. It was found that a small number of cumulative laser pulses per spot led to the formation of randomly distributed nanoparticles and HSFLs parallel to the incident laser polarization, while more accumulated pulses induced regular LSFLs perpendicular to the laser polarization. The transition mechanism was studied by real-time absorptance measurement and theoretical simulations. The results demonstrate that the interference between incident laser and SPPs excited by random surface scatterers facilitates prototype LSFL formation, which in turn enhances light absorption and SPP excitation for the following laser pulses. The effects of scanning velocity and laser fluence on LSFL quality were studied in detail. Large-area extremely regular LSFLs on ITO film with a diameter of 30 mm were efficiently fabricated by femtosecond laser direct writing with a cylindrical lens. The fabricated LSFLs on the ITO film demonstrated vivid structural color. During LSFL processing, a decrease in ITO film thickness leads to an increase in near-infrared optical transmittance.

## Figures and Tables

**Figure 1 materials-15-05092-f001:**
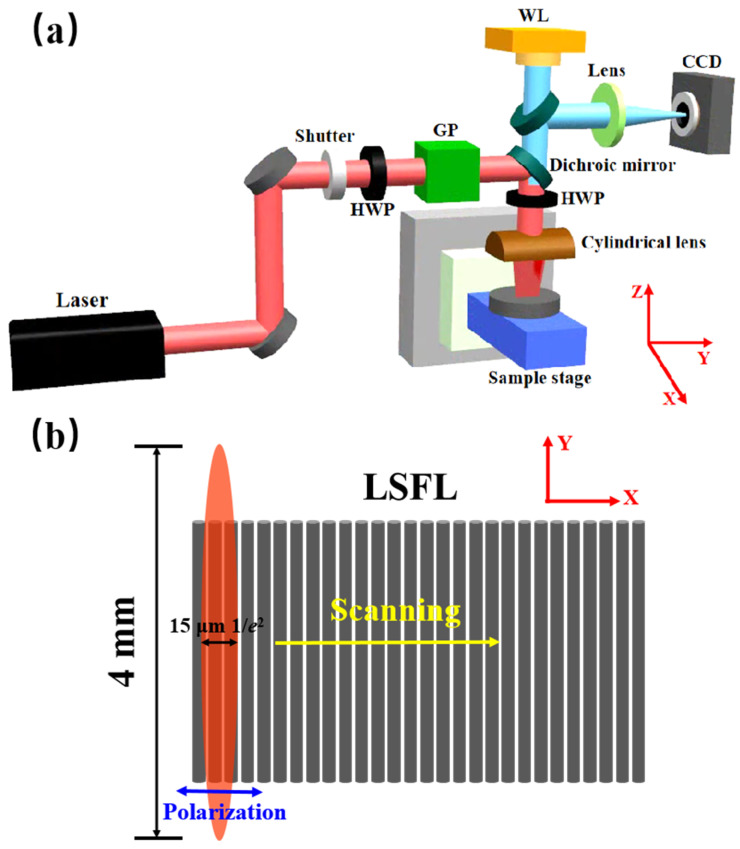
(**a**) Schematic of the laser direct writing setup. HWP is a half-wave plate, GP is a Glan prism and WL is a white light source. (**b**) Schematic of the laser polarization, focal spot and scanning direction.

**Figure 2 materials-15-05092-f002:**
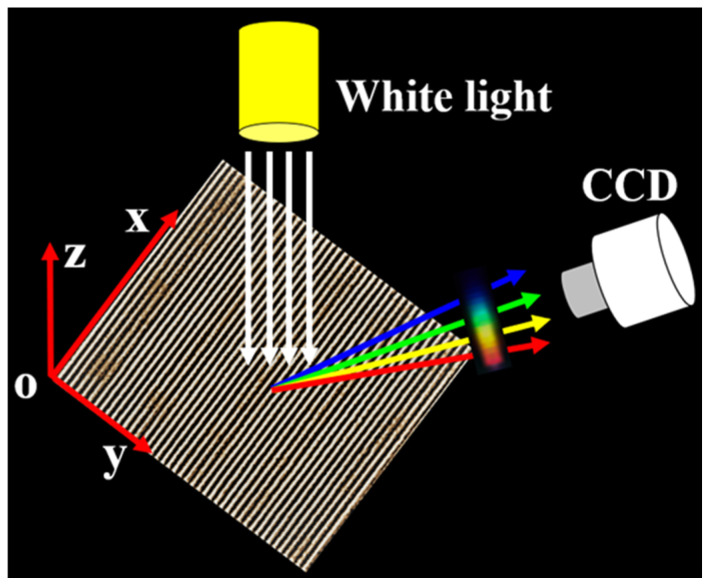
Schematic of the setup for structural color measurement.

**Figure 3 materials-15-05092-f003:**
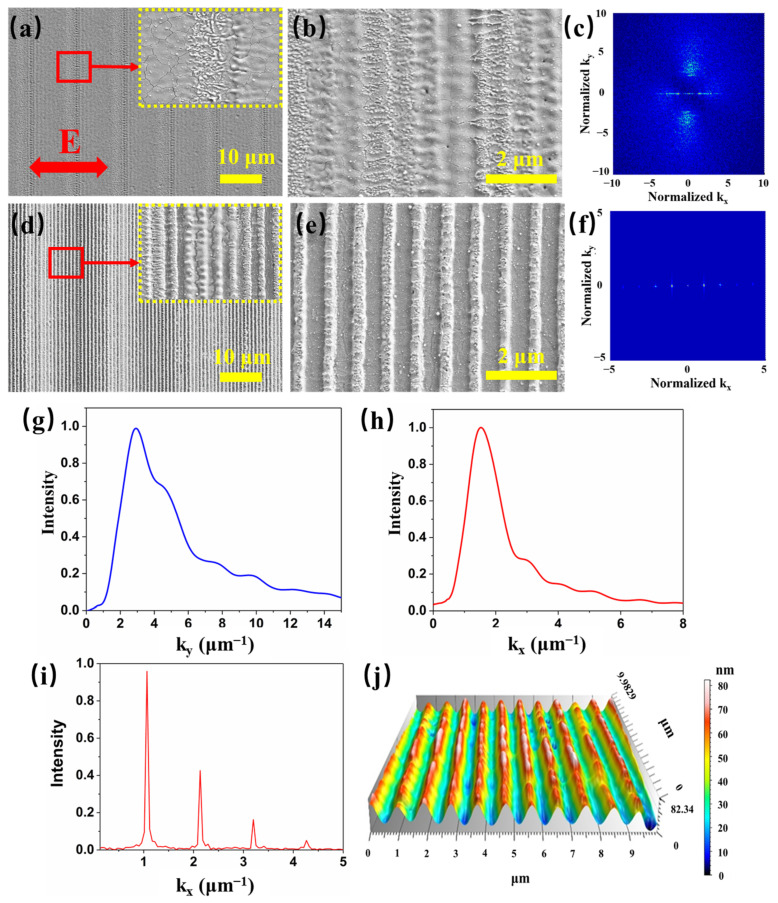
SEM images of micro/nanostructures on the ITO film after laser irradiation for scanning velocity of (**a**) 12, (**b**) 6, (**d**) 4 and (**e**) 3 mm/s with pulse fluence fixed at 0.66 J/cm^2^. The scale bars are 10 μm in (**a**) and (**d**) and 2 μm in (**b**) and (**e**). The insets in (**a**) and (**d**) are the enlarged SEM images for the red square areas. (**c**) and (**f**) are the 2D Fourier transform (FT) images of the LSFL in (**b**) and (**e**), respectively. (**g**) and (**h**) are the corresponding FT spectrums for k_x_ = 0 μm^−1^ and k_y_ = 0 μm^−1^ in (**c**), respectively. (**i**) is the corresponding FT spectrum for k_y_ = 0 μm^−1^ in (**f**). (**j**) is the confocal optical microscope image of (**e**).

**Figure 4 materials-15-05092-f004:**
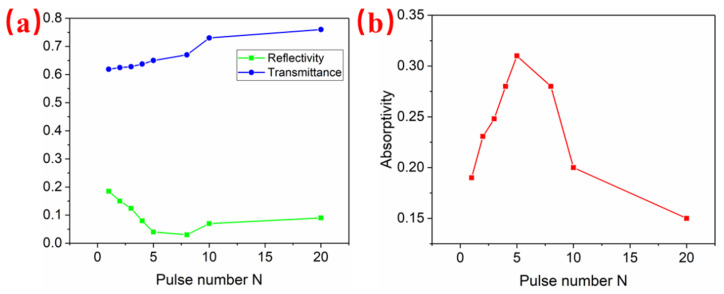
Measured (**a**) reflectance, transmittance and (**b**) absorptance of the ITO film for each consecutive femtosecond laser pulse. The pulse fluence was fixed at 0.66 J/cm^2^.

**Figure 5 materials-15-05092-f005:**
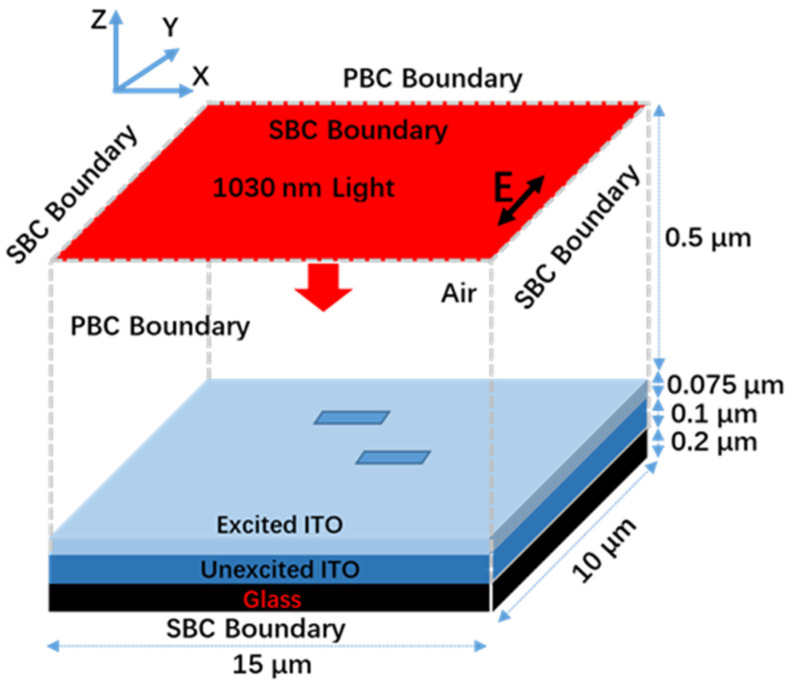
Schematic of the simulation setup with COMSOL Multiphysics software. The arrow E shows the laser polarization.

**Figure 6 materials-15-05092-f006:**
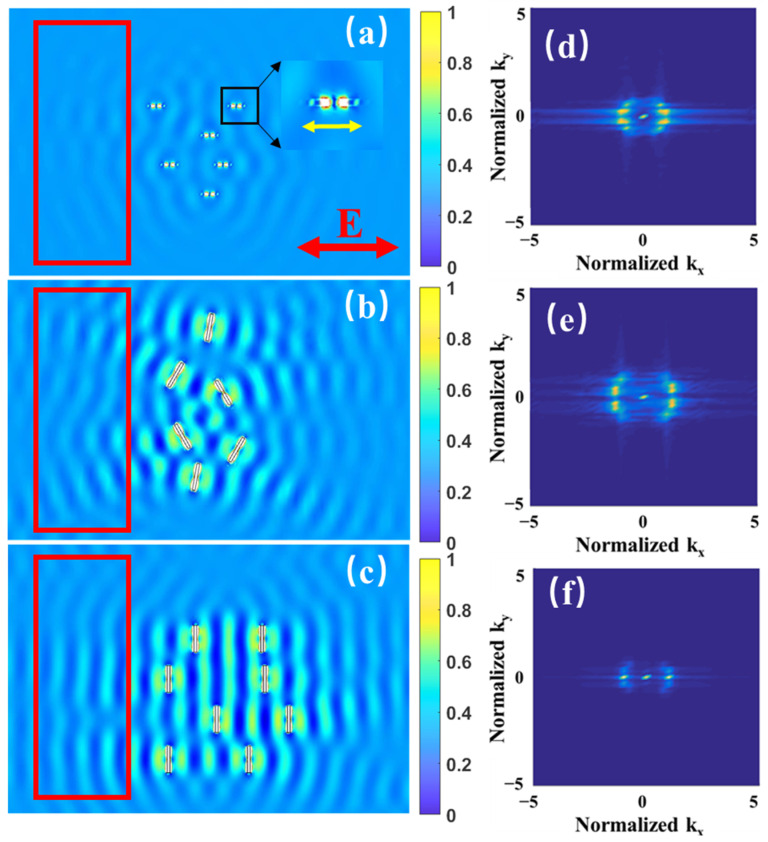
Surface light intensity distribution for different nanogroove orientations. Nanogrooves are (**a**) parallel to laser polarization (initial horizontal HSFLs), (**b**) mostly and (**c**) completely perpendicular to laser polarization (prototype LSFLs). (**d**–**f**) are the 2D FT images of the light field distributions in the red boxes of (**a**–**c**). The arrow E shows the laser polarization in (**a**). The color bars in (**a**–**c**) indicate the normalized light intensity.

**Figure 7 materials-15-05092-f007:**
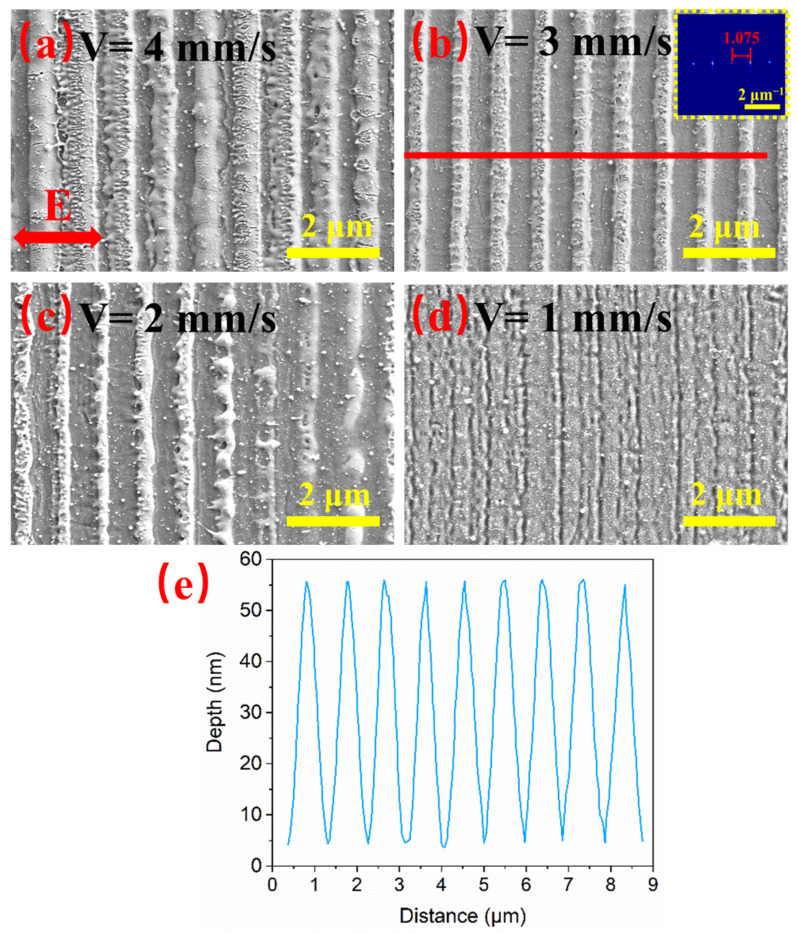
SEM images of LSFLs on the ITO film after laser irradiation under scanning velocities of (**a**) 4, (**b**) 3, (**c**) 2 and (**d**) 1 mm/s with a fixed pulse fluence of 0.66 J/cm^2^. The scale bars are all 2 μm. The FT of (**b**) is highlighted in the inset. (**e**) shows the depth profile along the red line in (**b**).

**Figure 8 materials-15-05092-f008:**
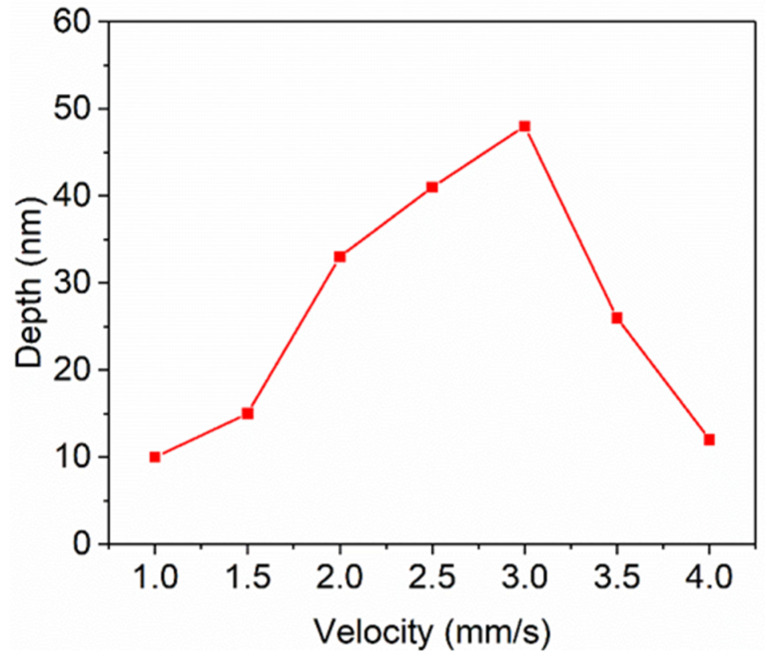
LSFL depth versus scanning velocity at a constant laser fluence of 0.66 J/cm^2^.

**Figure 9 materials-15-05092-f009:**
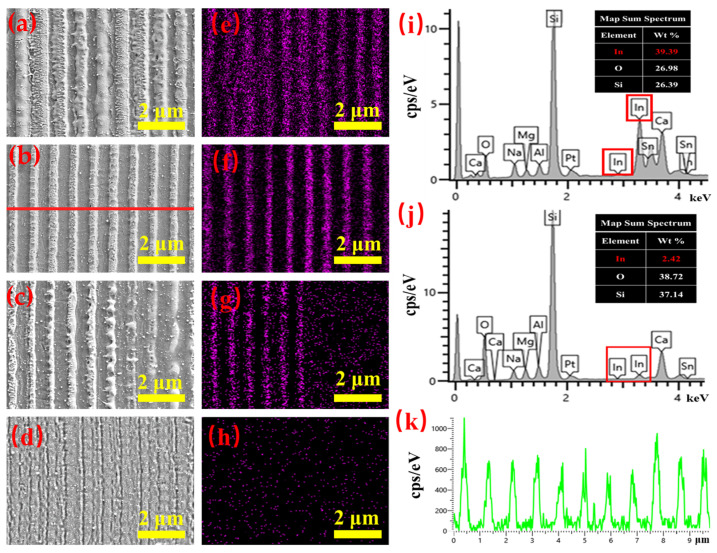
SEM images of LSFLs on the ITO film after laser irradiation under scanning velocities of (**a**) 4, (**b**) 3, (**c**) 2 and (**d**) 1 mm/s with fixed pulse fluence of 0.66 J/cm^2^. (**e**–**h**) Distributions of indium ion content measured by EDS corresponding to (**a**–**d**) respectively. The scale bars are all 2 μm. (**i**) and (**j**) are the sum spectra of (**a**,**d**), respectively. (**k**) is the distribution of the indium content along the red line in (**b**).

**Figure 10 materials-15-05092-f010:**
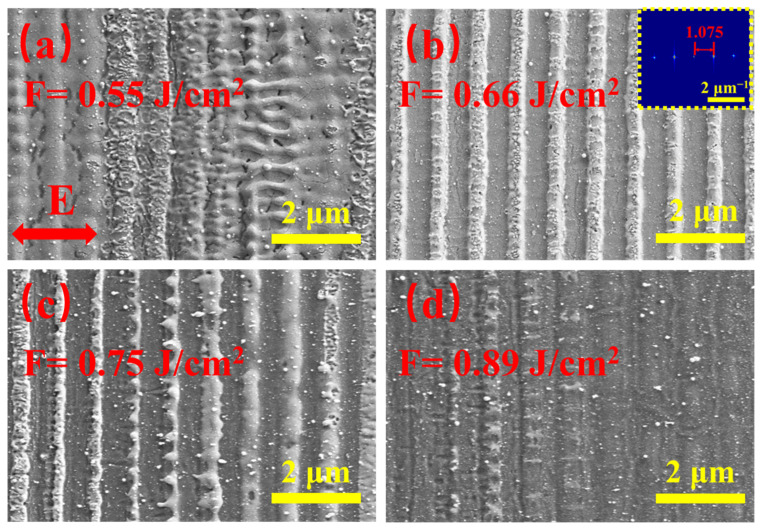
SEM images of LSFLa on the ITO film after laser irradiation under pulse fluences of (**a**) 0.55, (**b**) 0.66, (**c**) 0.75 and (**d**) 0.89 J/cm^2^ with a fixed scanning velocity of 3 mm/s. The scale bars are all 2 μm. The FT image of (**b**) is highlighted in the inset. The double arrow in (**a**) indicates the laser polarization direction.

**Figure 11 materials-15-05092-f011:**
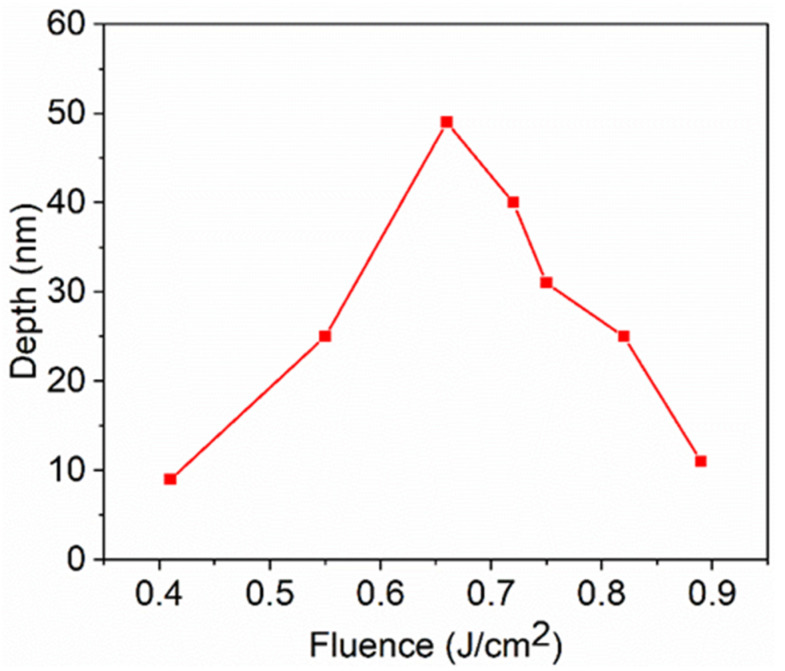
LSFL depth versus the pulse fluence at a constant scanning velocity of 3 mm/s.

**Figure 12 materials-15-05092-f012:**
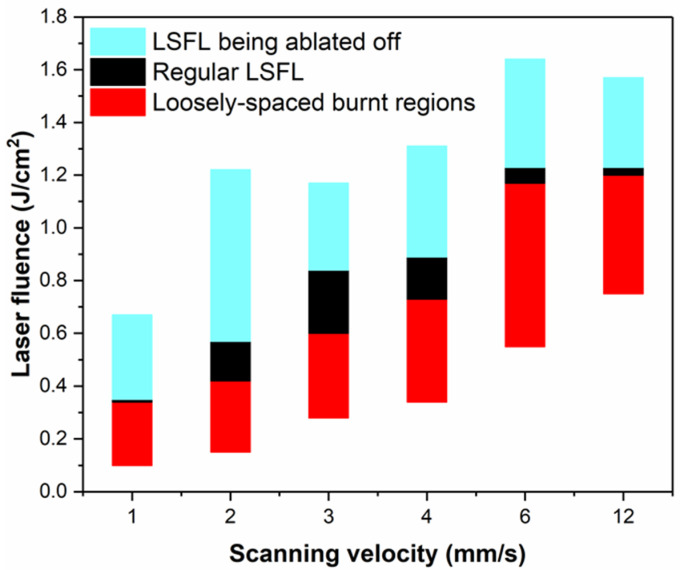
Different types of micro/nanostructures depending on the laser fluence and scanning velocity.

**Figure 13 materials-15-05092-f013:**
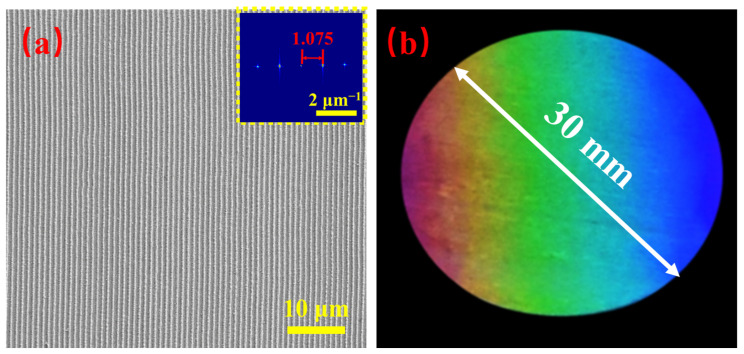
(**a**) SEM image of large-area regular LSFL. The scale bar is 10 μm and the inset is the 2DFT of the SEM image. (**b**) Optical image of the large-area LSFL.

**Figure 14 materials-15-05092-f014:**
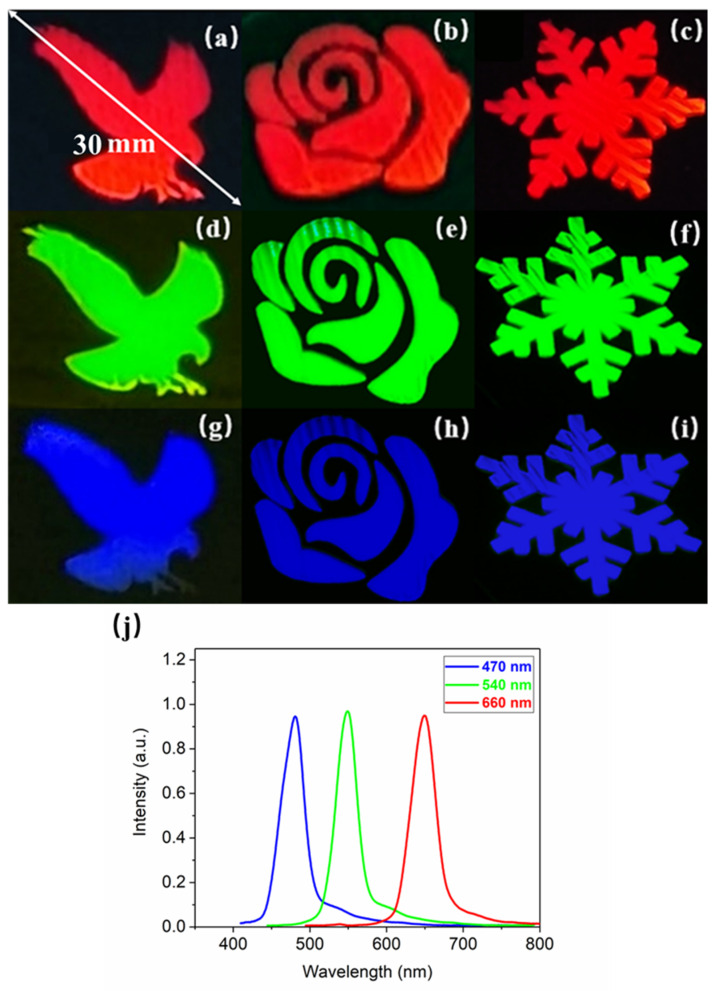
(**a**–**i**) Optical images, and (**j**) the diffraction spectra of different patterns composed of extremely regular LSFLs.

**Figure 15 materials-15-05092-f015:**
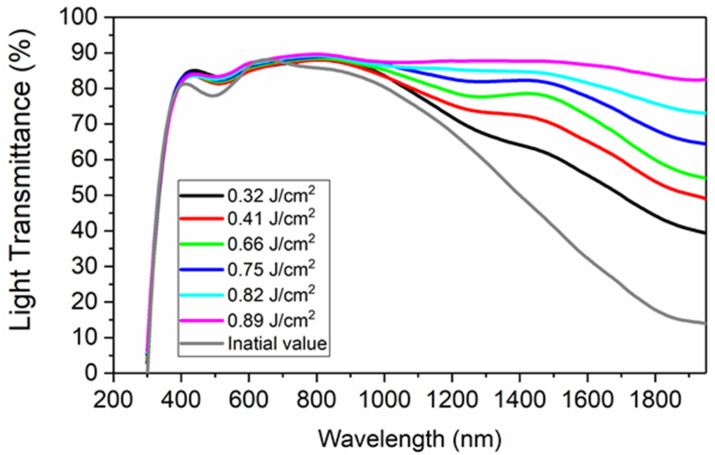
Light transmittance spectra of ITO films with LSFLs fabricated by femtosecond laser direct writing with different pulse fluences. The scanning velocity was fixed at 3.0 mm/s.

**Table 1 materials-15-05092-t001:** The average transmittance in the near-infrared band from 1200 to 1900 nm for LSFL fabricated by femtosecond laser direct writing with different pulse fluences (0~0.89 J/cm^2^).

Fluences (J/cm^2^)	0	0.32	0.41	0.66	0.75	0.82	0.89
Light transmittance (%)	37.96	56.83	65.30	71.32	76.91	81.32	90.96

## Data Availability

Not applicable.

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
