# Peer review of "Regular Periodic Surface Structures on Indium Tin Oxide Film Efficiently Fabricated by Femtosecond Laser Direct Writing with a Cylindrical Lens"

_materials, 2022, doi:10.3390/ma15155092_

Round 1
Reviewer 1 Report
Major
1. Section 3.1. In this paragraph the authors explain their proposal for the LIPSS formation mechanism. However, they should describe it more and provide some references to support their claims.
a. Lines 148-162. For the initial laser pulses, they state that the absorption is non uniform and random. This is only true when the pulse fluence is high enough to trigger point-deffect absorption and low enough so the multiphoton absorption is negligible. So this claims needs further experiments or a reference that positions the used fluence between these thresholds.
b. Lines 148-162. Then, they state that the initial features carved on the ITO surface serve to excite SPP with the same fluence as previously. This claim also needs to be supported by some numerical calculations of the field distribution. The authors did it in the following section, then, state it.
c. If I understand it correctly, the LIPSS of the SEM images were performed with the writing laser at normal incidence (not very clear in the manuscript). However, the study of the laser absorption were performed with a 70º incidence angle. It is known that the features of the LIPSS depend on the incidence angle (S Schwartz et al., JLMN-Journal of Laser Micro/Nanoengineering Vol. 13, No. 2, pp90, 2018; B Tan and K Venkatakrishnan 2006 J. Micromech. Microeng. 16 1080; Q. Wu et al., APPLIED PHYSICS LETTERS VOLUME 82, NUMBER 11, 2003 among many others). Authors should take into account the influence of the incidence angle on the LIPSS before comparing the absorption measurements with the SEM images of LIPSS fabricated with normal incidence.
2. Section 3.2.
a. Line 171: It is highly unclear how these ablation regions are distributed. Random number? Random position? Why 500x100x100 nm^3?
b. Modelled ITO thickness is 300 nm, while the samples used in experiments are ~175 nm thick. Why there is such a difference? Also, the optimum result almost ablates the totality of the ITO layer. In that case, a glass layer should be included in the model, and the influence of ITO layer thickness investigated.
3. Section 3.3. The authors show the results of some experiments varying the scanning velocity or pulse fluence around the optimal result.
a. However, there is no comment about what happens when they vary both magnitudes at the same time. Is it possible to find another pair of conditions which renders good results?
b. Authors state that it is important to ablate ITO almost to the point where it Also, how does the ITO layer thickness affect the production of LSFL?
4. Section 4.1, line 278. Authors state that the colors were very pure. They can support that claim with some bandwidth measurements.
5. Section 4.2. Authors state that the increase of NIR transmission is due to the LSFL. However, they also state that for the highest fluence, the transmission increases to 91% because the ITO layer was almost completely removed. So, in fact, there are to possible reasons for the transmission increase: the LSFL and the reduction of the layer thickness due to ablation. Authors must perform additional measurements to discriminate between them (for example, measuring the average thickness of the ITO layer after irradiation and compare it with the transmission of an unmodified layer of the same thickness).
6. Overall quality of the figures should be improved. Important data as bars distance, scanning velocities, etc should be noted in the figures. Also, false-color images should contain colorbars. Including all this information only in the figure caption makes very difficult to interpret them.
Minor
- * Figure 3d: A cross section profile would be desirable
- * Line 93: Better as an equation.
- * Lines 169-170: Reference to the data.
- * Table 1: Averaging the transmission results that vary so much in the 1200-1900 nm range seems a wrong thing to do.
- * Several typos
Author Response
Dear referee,
Thank you very much for reviewing our manuscript and giving us a chance to revise it. Your comments are very important for us to improve this manuscript. According to your suggestions, we made revisions in the resubmitted manuscript, as shown in the red words.
Please see the attachment. Our responses are in the attachment.

Reviewer 2 Report
This work is devoted to the formation of periodic surface nanostructures on the surface of an ITO film by femtosecond laser pulses using a cylindrical lens.
As the title of manuscript suggests, the use of a cylindrical lens is the key to this work. However, the text of the manuscript itself does not explicitly discuss the reasons for and purpose of using a cylindrical lens. Probably the goal is to have a larger area of ​​modification by stretching the beam waist in one of the directions, but this is just my guess. From this follows another question: how is the beam waist (the major axis of the ellipse) oriented with respect to the scanning direction?
As far as I understand, the main novelty of the study is to obtain structures with a long period on the surface of the ITO film, since the authors have a similar article (some of the pictures even match), where such structures are obtained on the surface of quartz glass. Then it is necessary to emphasize the advantages that are obtained in the case of ITO film.
Comments on the Experimental setup part.
1) What are the transverse dimensions of the beam waist? As I understand it, the minor axis of the ellipse is 15 microns. What is the size along the major axis of the ellipse? Rayleigh length obviously refers to the size of the waist along the direction of the laser beam, by definition.
2) How was scattering taken into account in the absorption measurement experiment?
3) Since the measurements of the diffraction grating were actually carried out, how were the losses due to higher diffraction orders taken into account?
Let's move on to the results and their discussion.
1) Figure 3 shows SEM images of LIPS obtained at different scanning speeds. How is the waist ellipse oriented with respect to the periodic structure? In which direction did the sample move during recording? The authors indicate that at a speed of 6 mm/s, HSFLs with a period of 330 nm are formed. What is the accuracy of determining the period? Based on Figure 3b, the period varies within certain limits, they must be indicated. The LSFL period also raises questions. The authors indicate a value of 660 nm, although it can be seen that the vertical lines coincide with the lines in fig. 3d, and it is already equal to 930 nm.
2) How were the permittivity values ​​used in the simulation obtained?
3) As I understand it, theoretical modeling confirms well the formation of periodicity in the direction perpendicular to the laser polarization. The phenomenon of the formation of HSFLs oriented parallel to the polarization remained unexplained.
4) Figure 7(e) shows the profile of the surface topography along the line shown in Figure 7(b). According by Figure 7(b), I would not expect sharp peaks on the profile, but flatter areas. Where did the sharp peaks come from?
5) In my opinion, the weakest point of the article is the statement about the increase in transmission in the IR region of the spectrum due to LSFL formation. The formation of LSFL can also be accompanied by ablation not only in the place of "depressions", but also on the rises, just a different degree of removal of materials. Here it is also necessary to add the transmission spectrum of the glass substrate on which the coating was applied. Most likely, the effect of increasing transmission in the IR region is associated simply with a gradual decrease in film thickness. To say that this is the effect of LSFL is incorrect.
There are a number of typos in the text:
1) In line 133, the dimension of the peak position on the Fourier spectrum should be µm^-1
2) Figure 4, x-axis labels should be corrected to "Pulse number N"
3) line 139, IFO film on ITO film
Author Response
Dear referee,
Thank you very much for reviewing our manuscript and giving us a chance to revise it. Your comments are very important for us to improve this manuscript. According to your suggestions, we made revisions in the resubmitted manuscript, as shown in the green words.
Please see the attachment. Our responses are in the attachment.

Reviewer 3 Report
The manuscript investigates the formation of LIPSS in ITO under fs laser pulse irradiation. It is clearly written and quite readable. The results are well presented. However, I have the objection that I do not see the novelty and significance of the findings. Please elaborate those (if there are any) more clearly. The formation of HSFL and LSFL in thin-film metals, semiconductors, including ITOs is well known for decades. The underlaying formation processes including the interaction of the laser light with surface plasmon polaritons is mostly accepted. Also influences of scanning speed and laser fluence are known. So, what is new? Before considering publication, the authors need to clarify the motivation and significance of the results including the advantages of using a cylindrical lens.
- The description of Fig 6 needs to be improved. What is seen in a,b,c? Where is the nanogroove orientation visible? What does the white / red colors mean? Are these the grooves?
Some minor points
Line 115: "was shown" should read "is shown"
Line 131 "was the" should read "depicts the"
Line 133 µ^-1
Caption of fig 4 "pluse" should read "pulse"
Line 139 "would be" should read "is"
Line 168 "state" should read "states"
Line 250/fig 11 Is the axial resolution of the confocal microscope really sufficient to resolve 10 nm differences?
Author Response
Dear referee,
Thank you very much for reviewing our manuscript and giving us a chance to revise it. Your comments are very important for us to improve this manuscript. According to your suggestions, we made revisions in the resubmitted manuscript, as shown in the blue words.
Please see the attachment. Our responses are in the attachment.

Round 2
Reviewer 1 Report
Authors have correctly answered all my concerns/suggestions in the previous report and made the correct changes in the manuscript, except for one. Authors state in lines 380-382 of the new manuscript that the increase of transmittance s mainly due to the reduction of ITO layer thickness (which means that the nanostructures also contribute to this reduction). Then, in their answer, they provide additional data: transmittance of an unmodified layer of 25 nm thickness (~73%). They also provide an estimation of average layer thickness after irradiating with 0.66 J/cm of 24 nm, which produces an average transmission of ~71%. This is lower than the untreated case, so the authors should remove their claim that the nanostructures increase the transmissivity from section 4.3 and the conclusions, as with the available data, the only claim they can make is that it reduces it.
Reviewer 2 Report
Thank you for made corrections.
1) In the same time, in several parts of manuscript text should be corrected: "The fabricated LSFL on the ITO film demonstrate ... increased transmittance in the near-infrared band". The authors wrote the reason of increased transmittance was decrease of thickness of film, not due LSFL itself. This statement should be pointed out in abstract and conclusion.
2) About fig 7. What is the reason of so drastic difference between SEM image and confocal image of surface morphology? In SEM image flat patterns are clearly seen, but in confocal image surface height changes as sin-like function.
